# Positioning System of Infrared Sensors Based on ZnO Thin Film

**DOI:** 10.3390/s23156818

**Published:** 2023-07-31

**Authors:** Chia-Yu Tsai, Yan-Wen Lin, Hong-Ming Ku, Chia-Yen Lee

**Affiliations:** 1Department of Materials Engineering, National Pingtung University of Science and Technology, Pingtung 912, Taiwan; years39191@gmail.com (C.-Y.T.); yanwen881126@gmail.com (Y.-W.L.); 2Department of Chemical Engineering, King Mongkut’s University of Technology Thonburi, Bangkok 10140, Thailand; hongming.ku@gmail.com

**Keywords:** MEMS, positioning system, pyroelectric infrared sensors, zinc oxide

## Abstract

Infrared sensors incorporating suspended zinc oxide (ZnO) pyroelectric films and thermally insulated silicon substrates are fabricated using conventional MEMS-based thin-film deposition, photolithography, and etching techniques. The responsivity of the pyroelectric film is improved via annealing at 500 °C for 4 h. The voltage response of the fabricated sensors is evaluated experimentally for a substrate thickness of 1 µm over a sensing range of 30 cm. The results show that the voltage signal varies as an inverse exponential function of the distance. A positioning system based on three infrared sensors is implemented in LabVIEW. It is shown that the position estimates obtained using the proposed system are in excellent agreement with the actual locations. In general, the results presented in this study provide a useful source of reference for the further development of MEMS-based pyroelectric infrared sensors.

## 1. Introduction

Infrared sensors, which consist of a pyroelectric thin film sandwiched between two metallic electrodes, produce an electrical voltage (or current) in response to temperature variations. In recent decades, they have been widely used in fire detection, intruder alarms, and infrared light detection, as well as imaging, body temperature monitoring, respiration, and heart rate detection, and so on [1,2,3,4,5]. Moreover, pyroelectric sensors can be easily integrated with modern IC circuits to create sophisticated system-on-chip (SoC) devices for a wide variety of sensing and detection applications [6,7].

Lienhard and Heepmann [8] investigated the transmittance of thin nickel (Ni) films deposited on pyroelectric polyvinylidene fluoride (PVDF) films over the infrared spectral range of 2–50 μm. It was shown that an appropriate specification of the Ni film thickness could effectively control the absorbance characteristics of the pyroelectric sensor. Chong et al. [9] fabricated a thin-film pyroelectric sensor array consisting of 16 sensing elements made of metal–ZnO–metal sandwiched layers with dimensions of 200 μm × 200 μm. The sensor response was enhanced to have a maximum voltage sensitivity of 110 mV/mW by depositing the sensing elements on a thermally isolated and freestanding Si_3_N_4_/SiO_2_ membrane at a cutoff frequency of 20 Hz. The membrane was created by removing the underside of the SiO_2_ substrate using a conventional back-etching technique. Wei et al. [10] fabricated ZnO pyroelectric sensors with partially covered and fully covered electrodes, respectively, for performance comparison. The experimental results showed that the responsivity of the partially covered device was around four times higher than that of the fully covered device for incident light frequencies in the range of 10–1000 Hz.

Hsiao et al. [11] fabricated a ZnO pyroelectric sensor on a back-etched silicon wafer using a two-step radio frequency (RF) sputtering technique. A ZnO film with preferred *c*-axis orientation and an enhanced voltage response could be found via the use of a two-step deposition process. The sensor achieved a maximum voltage sensitivity of 8.6 mV/mW at a cutoff frequency of 500 Hz when implemented with a Ni coating on the uncovered part of the ZnO film, in order to enhance the absorption of the incident energy. Hsiao and Yu [12] enhanced the response of pyroelectric devices by increasing the temperature variation rate. The aerosol deposited (AZ) ZnO film was further treated by CO_2_ laser annealing with a wavelength of 10.6 µm, a power of 25 W, and a beam diameter of 2 mm in a N_2_ atmosphere. Moreover, at a high frequency of 3000 Hz, the ZnO pyroelectric device with comb-like electrodes possessed a voltage response four times greater than that with a fully covered electrode. Hsiao et al. [13] developed a pyroelectric sensor consisting of a zinc oxide (ZnO) sensing layer, gold (Au) upper and lower electrodes, and a supporting silicon (Si) substrate. The response voltage of the device was measured for three different line widths of the upper electrode, given different values of the thin film thickness. The results showed that the optimal electrode width decreased as the thickness of the ZnO sensing film reduced. The maximum voltage response was obtained for a line width of 30 μm and ZnO thickness of 1µm. Cieck et al. [14] deposited ZnO thin films by RF&DC magnetron sputtering on silicon and porous silicon (PS) substrates for pyroelectric applications. Due to PS’s large internal area, low thermal conductivity, and preferred *c*-axis orientation, a pyroelectric coefficient of 8.2 could be achieved (about 40 times higher than that on silicon substrate).

Hsiao et al. [15] fabricated multi-frequency band pyroelectric sensors with various ZnO layer thicknesses of 0.8 µm, 6 µm, 10 µm, and 16 µm for detecting subjects with various velocities. The experimental results showed that increasing the thickness of ZnO layer could compensate for the tardy response at a high-frequency band. Sharmila et al. [16] proposed using an optical sensing and annealing effect on RF-sputtering ZnO thin films. It was observed that the ZnO layer with 200 °C and 300 °C annealing temperatures showed improved UV detection properties in comparison to 100 °C. The ZnO layer with a higher annealing temperature performs better in the field of UV photosensing applications. Lee [17] fabricated a pyroelectric infrared sensor incorporating a ZnO sensing layer, a thermally insulated Si substrate, and Au upper and lower electrodes. The backside of the wafer was selectively etched to create a suspended sensing structure. Moreover, the responsivity of the device was enhanced by annealing the thin film at 500 °C for four hours. The numerical and experimental results showed that the voltage responsivity of the proposed device improved as the substrate thickness reduced.

One of the most common applications of infrared sensors is that of position sensing. Kim [18] proposed an indoor positioning system consisting of three infrared emitters (scanner) placed at known positions and a mobile incident angle sensor attached to a moving target to measure the angle differences between each pair of emitters. Gorostiza et al. [19] proposed a method for achieving the accurate localization of mobile robots in intelligent spaces by sensing the infrared signals emitted by the robots and then applying a hyperbolic trilateration method to the corresponding differential distances. Luo et al. [20] constructed a wireless sensor network consisting of pyroelectric infrared sensor arrays as nodes and processed the signals emitted by the sensors using a filtering algorithm to detect human motion and perform activity recognition. Yan et al. [21] investigated the various macro and micro factors affecting the ability of pyroelectric sensor systems to perform human identification. The results showed that the recognition performance improved with a closer distance, a faster moving speed, and a greater number of signal modulation mask holes. Agmell et al. [22] constructed an indoor localization system for AGVs based on mobile receivers and infrared scanners mounted on the wall. Zhang et al. [23] proposed a low-cost and robust method combining infrared vision and ultra-wideband (UWB) fusion for the indoor localization of mobile robots. It was shown that a centimeter level of positioning accuracy could be obtained by applying an adaptive weight positioning algorithm, to detect the edge of the UWB coverage area, and fusing the outputs of the two sensing mechanisms using an extended Kalman filter (EKF). Wu et al. [24] presented pyroelectric detection and sensing signal processing algorithms to create a non-wearable cooperative indoor human localization system based on pyroelectric sensor networks.

The present study proposes an approach for performing target object localization in indoor environments using a trilateration approach based on three miniaturized ZnO-based pyroelectric infrared sensors. The voltage readings of the three sensors are interfaced to a computer through low-noise current preamplifiers, and the signals are then fused and processed using a self-written LabVIEW human–machine interface to determine the position of the target object in two-dimensional space.

## 2. Principles and Designs

The pyroelectric phenomenon is primarily generated by the influence of the environmental temperature on the charged substances within the crystal structure of the sensing material. Assume that infrared light with a power *W(t)*, sinusoidally modulated at a frequency ω, is incident on the surface of a pyroelectric sensing element with a pyroelectric coefficient *p*, an electrode area *A*, thickness *d*, and surrounding temperature *T*. The incident power can be formulated mathematically as follows [25]:(1)$$W\left(t\right)=W_{0}e^{iwt}$$
where *i* = *ApdT*/*dt*.

Assume further that the sensing element has a thermal capacity *H_T_*, and the thermal conductance to the surroundings is denoted as *G_T_*, giving a thermal time constant of $\tau_{T}=H_{T}/G_{T}$. Given an emissivity η, the temperature difference between the element and its surroundings, θ, can be described as [25]:(2)$$\eta W\left(t\right)=H_{T}\frac{d\theta }{dt}+G_{T}\theta$$
which has the solution [25]:(3)$$\theta =\frac{\eta W_{0}e^{iet}}{G_{T}+i\omega H_{T}}$$

Differentiating Equation (3), the rate of the temperature difference between the element and its surrounding is obtained as:(4)$$\frac{d\theta }{dt}=\frac{\omega \eta W_{0}}{G_{T}{\left(1+\omega^{2}\tau_{T}^{2}\right)}^{\frac{1}{2}}}$$

In particular, a change in the temperature induces a stress variation within the sensing layer, which is manifested by a measurable change in the electrical charge and output current of the device. The pyroelectric variation rate, $∆P_{S}$, can be expressed as follows:(5)$$∆P_{S}=p\times ∆\theta$$
where ∆θ is the temperature change. Figure 1 illustrates the basic operation of a pyroelectric sensor with a surface area A and pyroelectric coefficient component *p*’ in the vertical direction. When exposed to an input energy flux, the pyroelectric current, which flows through the external circuit, is given by
(6)$$i_{p}=A\times \frac{dP_{S}}{dT}=Ap^{\prime }\times \frac{d\theta }{dt}$$

It can be found that the pyroelectric sensor is “AC coupled” to any input energy flux that generates a change in temperature, where this input may primarily take the form of electromagnetic radiation absorbed in the pyroelectric material.

Figure 2a presents a structural assembly diagram of the thin-film pyroelectric infrared sensor proposed in the present study based on a single-side polished Si wafer substrate with a thickness of 525 ± 25 μm. As shown, the substrate is coated on both sides with a silicon nitride (Si_3_N_4_) insulating layer with a thickness of 1 μm. The bottom and top electrodes each comprise a layer of Cr with a thickness of 0.03 μm and a layer of Au with a thickness of 0.1 μm, and have the configurations and dimensions shown in Figure 2b,c, respectively. Finally, the sensing layer has the form of a 0.89-μm thick ZnO film sandwiched between the top and bottom electrodes and has the dimensions shown in Figure 2d.

Figure 3 shows the basic steps in the self-written LabVIEW program used to perform target object detection in the present study. Upon receiving the initial voltage signal of the pyroelectric sensor in the absence of an infrared target object, the program measures the voltage and displays it on a human–machine interface (HMI). Following the introduction of the target object into the sensing field, the program measures the voltage once again, calculates the voltage change, and presents it on the interface. The voltage change is then input to the MathScript module in LabVIEW to determine the position of the object using a triangulation-based method. Finally, the computed coordinates of the target object in the two-dimensional (2D) sensing field are displayed on the HMI.

## 3. Methods and Fabrication

Fabrication

Figure 4 shows the main steps in the sensor fabrication process. The process commenced by patterning the bottom electrode. A low-pressure chemical vapor deposition (LPCVD) method was first employed to deposit a 1-μm thick Si_3_N_4_ film on both sides of a single-side polished P-type silicon wafer substrate as an insulating layer. The deposition process was performed using high-purity chemical gases with a chamber pressure of less than 10 Torr. A 300-Å thick Cr adhesive layer and 1000-Å thick Au conducting layer were then deposited on one side of the substrate using an electron beam evaporation technique with a chamber pressure of less than 100 Torr to form the bottom electrode. The electrode was patterned using a standard photolithography method.

Having patterned the lower electrode, a ZnO sensing layer was deposited on the electrode using a high-vacuum magnetron sputtering system with a 4N purity grade 4” ZnO target. The wafer was placed in the chamber, and the chamber was vacuumized down to 5 × 10^−6^ Torr. A pre-sputtering process was performed using argon (Ar) gas to remove any oxides or impurities from the target surface. In the pre-sputtering process, the target power was increased at a rate of 10 W per minute until it reached 150 W, and the target was then bombarded with Ar ions for 30 min with an Ar inlet flow rate of 30 sccm. The total sputtering time is 2 h with a heated substrate temperature of 220 °C.

The upper electrode was patterned using photoresist, and a 300-Å thick Cr layer and 1000-Å thick Au layer were deposited using an electron beam evaporation method. The photoresist layer was removed using acetone solution to leave behind just the desired pattern of the top electrode.

In the final step of the fabrication process, a backside etching technique, was used to release the sensing structure. A photolithography process was first performed to define the etching area on the backside of the wafer. Reactive ion etching (RIE) was used to remove the 1 µm thick Si_3_N_4_ insulation layer from the wafer surface. The flow rates of CF_4_ and O_2_ are 25 and 5 sccm, respectively. The total etching time was 12 min with a power of 100 W. Wet etching was then performed in a 30 wt% potassium hydroxide (KOH) aqueous solution at a temperature of 80 °C for 6 h to produce a final suspended film thickness of 1 μm. The average wet etching rate was 1.5 µm/min [17].

As shown in Figure 5, sixteen infrared sensing chips were fabricated on a single silicon wafer. Four square alignment marks located at the four corners of the photomask were used to ensure a precise alignment of the mask during the patterning process. Figure 6 presents a photograph of the final sensor.

## 4. Results and Discussion

This section commences by analyzing the structure and surface morphology of the ZnO pyroelectric sensing layer. The voltage response of the device is then investigated as a function of the distance from the target object. Finally, the feasibility of the proposed sensor for target object detection and localization is demonstrated.

The ZnO film was sputtered using pure argon (Ar) as the working gas and was then annealed at 500 °C for 4 h. The structure of the annealed ZnO film was examined by X-ray diffraction (XRD) using monochromatic Cu-Kα with a wavelength of λ = 0.154 nm as the excitation source. Scanning was performed over a range of 2*θ* = 30~60°, with a sampling interval of 0.03° and a sampling time of 0.45 s. The analysis was conducted using an operating voltage of 40 kV and an operating current of 40 mA.

The XRD pattern for the annealed ZnO film is shown in Figure 7. The results indicate that the strongest peak is associated with the (002) crystal orientation. Thus, the ZnO film is inferred to have a wurtzite structure [18].

Figure 8a,b presents SEM images of the surface morphologies of the as-deposited and annealed ZnO thin films, respectively. The as-deposited surface has a flake-like morphology with relatively large gaps and voids between the grains. Following the annealing process, the grain size increases while the voids and gaps decease in both size and number, and the crystal structure becomes even denser. As a result, the thermal conduction performance of the ZnO sensing layer improves, leading to enhancement of the pyroelectric current signal transmission capability, as discussed in [17].

Figure 9 illustrates the basic structure of the measurement platform constructed in the present study. The sensor was placed on an X-Y platform and a human palm was utilized as the thermal source for infrared signals. The sensor signal was interfaced to an SR570 low-noise current preamplifier (Stanford Research Systems) via SMA connectors, and the resulting voltage signal was displayed in real time on an oscilloscope (TDS 2001C, Tektronix, Beaverton, OR, USA).

Figure 10 shows the sensing experimental results obtained for the voltage responsivities of the fabricated ZnO pyroelectric sensors at cutoff frequencies in the range of 0–1800 Hz. From inspection, the maximum voltage response is equal to 770 mV and is obtained at a cutoff frequency of 1000 Hz.

Figure 11 presents the experimental results obtained for the variation of the voltage change of the infrared sensor with the square of the distance between the sensor and the human palm. It can be found that the voltage response is 160 mV, as the sensor is normally located just beyond the light source with nearly 0 cm of distance. As expected, the voltage response decreases as the sensing distance increases. At a distance of 30 cm, the voltage response decreases to be 1.6 mV, which means that the upper sensing limit of the developed sensor is 30 cm. The voltage change is related to the square of the sensing distance inversely and exponentially, with a correlation coefficient of R^2^ = 0.9885. The exponential relationship experimentally fits in with Equation (1). Moreover, the voltage response is particularly pronounced for sensing distances in the range of 0~30 cm.

Figure 12 shows the measured voltage change of the sensor over a distance of 0~30 cm as the position of the human palm relative to the sensor was rotated through 180°. It is seen that the voltage variation is identical at equidistant points over the range of 120° in front of the sensor (as indicated by the blue region). Hence, in developing the target object positioning system proposed in this study, the sensing angle range of each sensor was set as 120°. Within this range, the voltage varied from 1.6 mV at a distance of 30 cm to 160 mV at a distance of 0.1 cm.

A target object detection system was constructed using a system of three sensors. For each sensor, the acquired data were converted into a 1D array of scalars, representing a single channel of voltage values. Due to slight fluctuations of the voltage signal, 10 data points were averaged to obtain a mean value when determining the initial voltage (no target object) and measurement voltage (with target object) of the sensor, respectively. Figure 13a,b shows the code used to measure the initial voltage of the three individual sensors and the voltage changes following the introduction of the infrared source into the system, respectively.

The three sensors were connected to the SR570 low-noise current preamplifier using SMA connectors. The amplified voltage signals were interfaced to LabVIEW software (Community Edition) via an NI-9234 DAQ module. The positioning experiments were conducted using a high-power infrared source as the target object (Figure 14). The voltage signals of the three sensors were first measured with the light turned off. The light was then turned on, and the voltage signals of the three sensors were measured once again. The precise position of the light source was determined using a trilateration approach based on the three voltage change measurements (Figure 15). The trilateration approach is a measurement method based on the detected distances of the three sensors overlapping triangles on the platform of the experiment for the determination of the relative position of the light source. The coordinate of the light source can be determined by solving the three simultaneous circle equations derived from the performance curve (Figure 11) of the three sensors on the same coordinate system of the same platform. Finally, the coordinates of the light source were displayed as a plotted point on an X-Y graph presented through the LabVIEW HMI (Figure 16).

Table 1 compares the measured coordinates of the infrared light source with the computed coordinates for three positioning tests. As shown, the average positioning error lies within ± 0.13 cm. Within the 40 × 20 cm^2^ of experimental platform, the mean accuracy is 99%, which is above those of the previous studies of indoor positioning systems based on infrared sensors [19,20,23,24]. Though high positioning accuracy can be attained within a small area (40 × 20 cm^2^), this study proposes a straightforward algorithm to measure the coordinate of an IR light source with a simpler configuration.

An indoor positioning system is a system that can continuously determine the position of an object in real time, but indoor environments are normally full of multiple objects and obstacles that reflect signals, which lead to signal fluctuation and noise issues. Furthermore, indoor position suffers from interference from surrounding systems, such as wireless and mobile devices, that can cause signal instability and signal strength fluctuations [22]. In addition to the enhancement of the working distance of the proposed sensors, more signal processing functions should be implemented in the positioning system for their future applications.

## 5. Conclusions

This study has presented a MEMS-based ZnO thin-film infrared sensor for non-contact temperature and distance sensing. The XRD results have shown that, following annealing at 500 °C for four hours, the ZnO film demonstrates a strong peak in the (002) crystallographic direction, indicating that it has a wurtzite structure. It has been shown that the ZnO sensor can function as an infrared-based target detection system over the range of 0~30 cm. A trilateration-based object localization system has been constructed based on three sensors and a self-written LabView interface. The experimental results have shown that the estimated target position deviates from the actual position by no more than 0.13 cm.

## Figures and Tables

**Figure 1 sensors-23-06818-f001:**
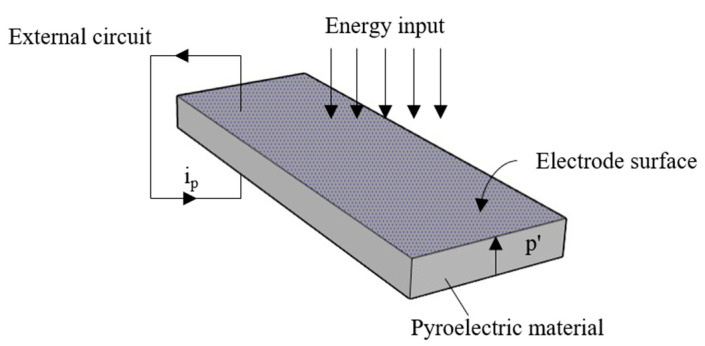
Electricity generation of pyroelectric sensor in response to temperature change [17].

**Figure 2 sensors-23-06818-f002:**
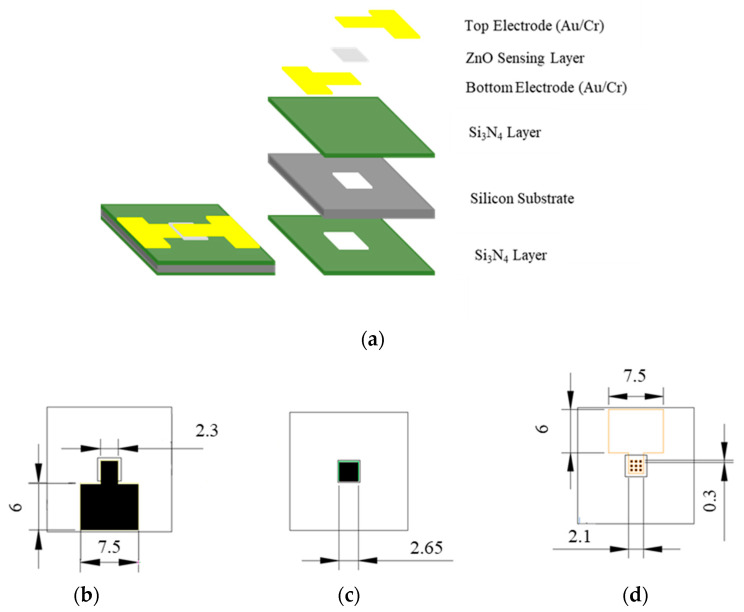
Structural assembly and component details of proposed MEMS-based pyroelectric infrared sensor: (**a**) structural arrangement of sensor; dimensions and configurations of: (**b**) bottom electrode, (**c**) top electrode, and (**d**) sensing layer. Unit: mm.

**Figure 3 sensors-23-06818-f003:**
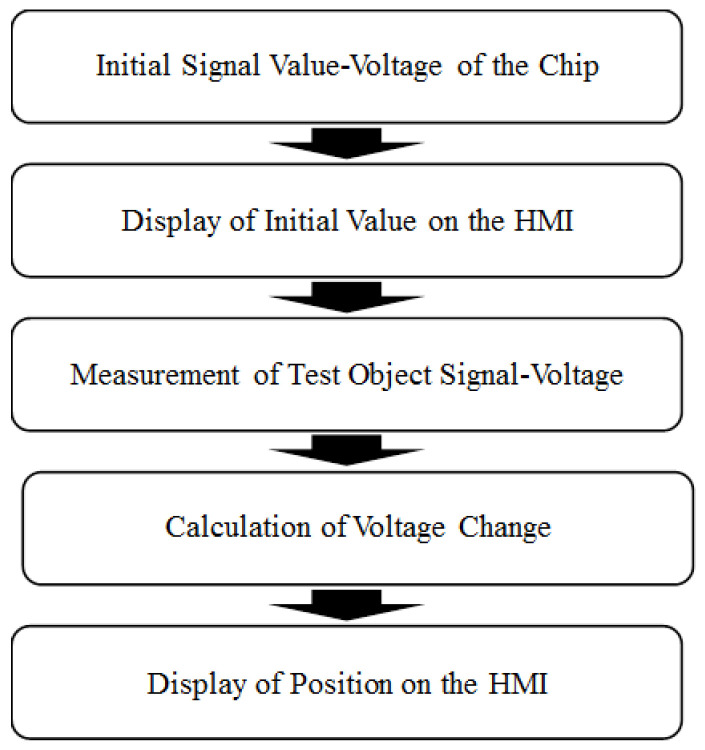
Flowchart showing main steps in proposed localization system.

**Figure 4 sensors-23-06818-f004:**
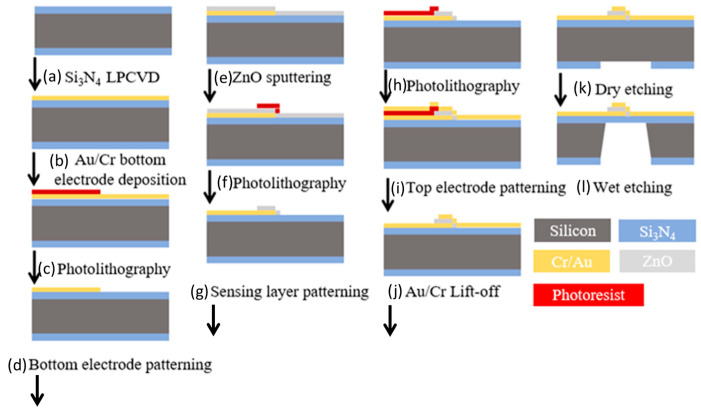
Process flowchart showing main steps in sensor fabrication process: (**a**) LPCVD deposition of Si_3_N_4_, (**b**) E-beam deposition of Au/Cr bottom electrode, (**c**) photolithography, (**d**) patterning of bottom electrode by wet etching, (**e**) RF sputter deposition of ZnO thin film, (**f**) photolithography, (**g**) patterning of sensing layer by wet etching, (**h**) photolithography, (**i**) E-beam deposition of Au/Cr top electrode, (**j**) lift-off, (**k**) dry etching of back Si_3_N_4_ layer, and (**l**) KOH wet etching of silicon substrate [17].

**Figure 5 sensors-23-06818-f005:**
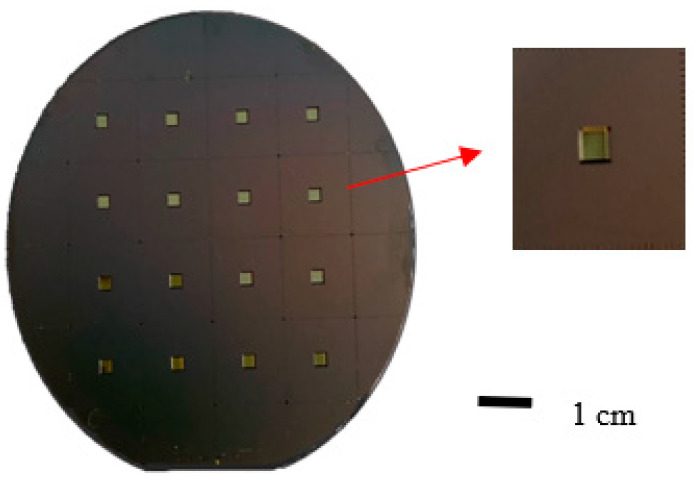
Photographs of back-etched chip.

**Figure 6 sensors-23-06818-f006:**
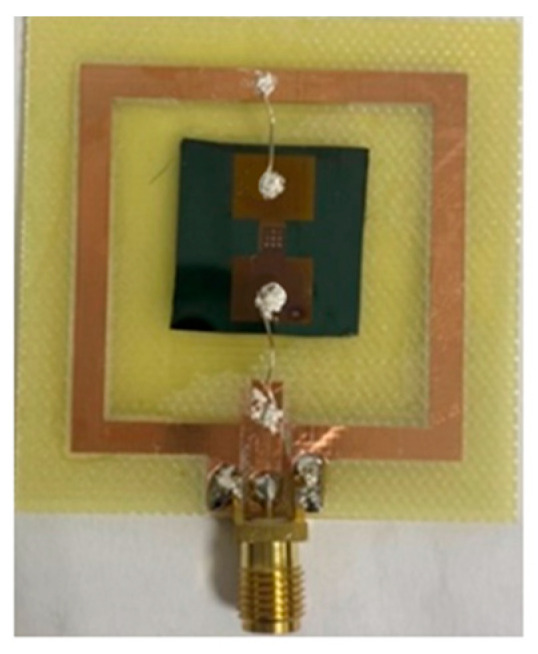
Photograph of finished pyroelectric infrared sensor.

**Figure 7 sensors-23-06818-f007:**
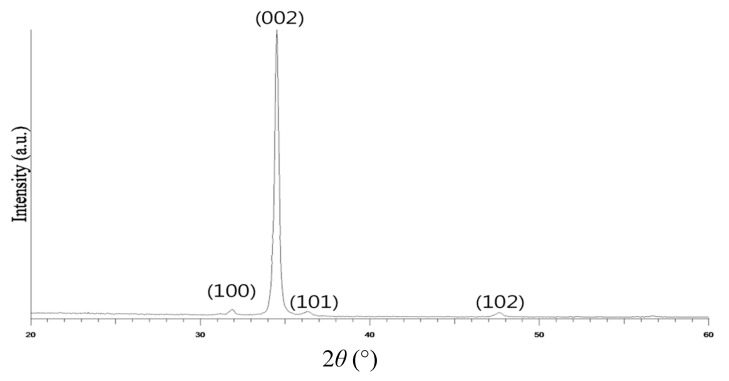
XRD pattern of ZnO sensing film.

**Figure 8 sensors-23-06818-f008:**
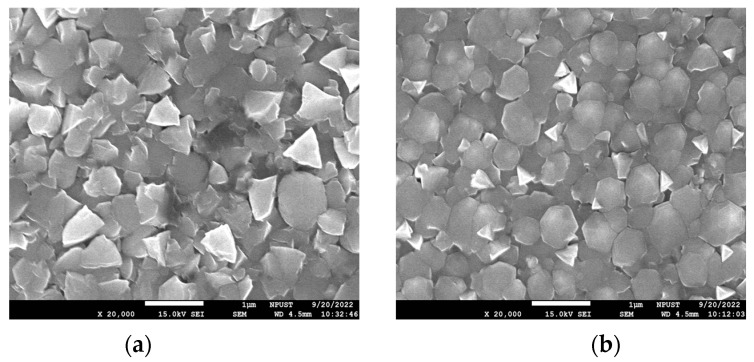
SEM morphologies of ZnO sensing layers: (**a**) as-deposited ZnO film, and (**b**) ZnO film annealed at 500 °C for 4 h. Magnification: ×20,000.

**Figure 9 sensors-23-06818-f009:**
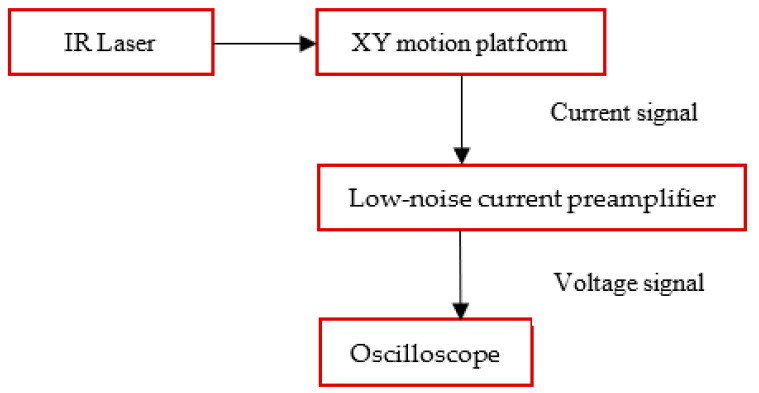
Schematic illustration of experimental measurement platform.

**Figure 10 sensors-23-06818-f010:**
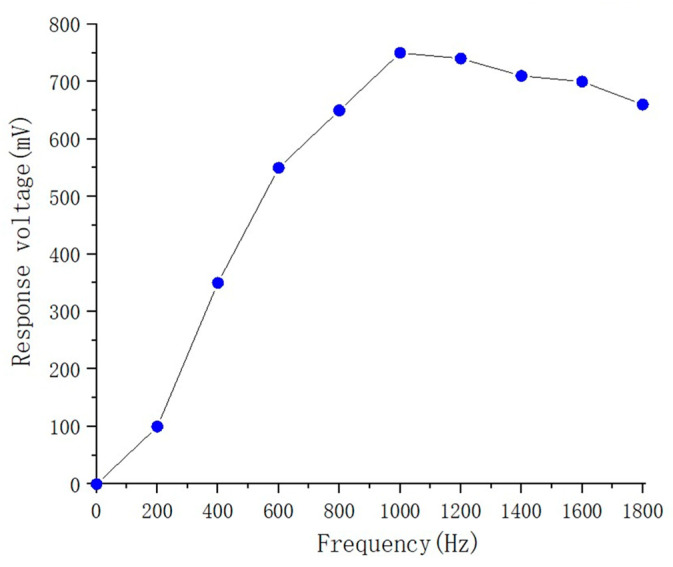
Voltage responsivity of fabricated pyroelectric infrared sensor with cutoff frequencies in the range of 0–1800 Hz.

**Figure 11 sensors-23-06818-f011:**
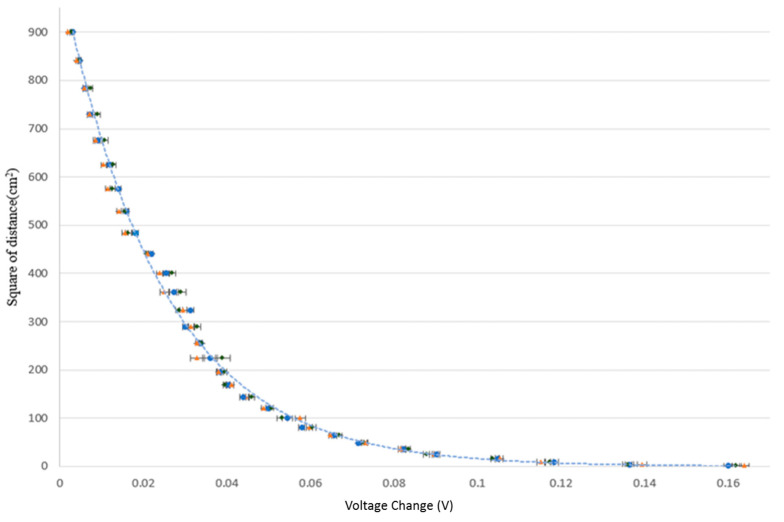
Voltage response of infrared sensor as function of sensing distance.

**Figure 12 sensors-23-06818-f012:**
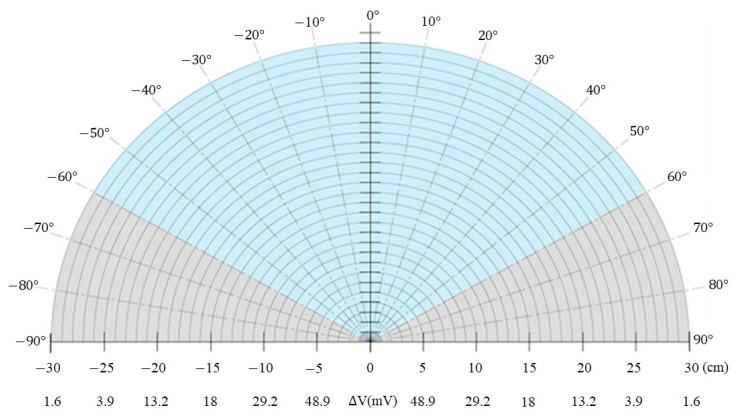
Angular measurement range of infrared sensor.

**Figure 13 sensors-23-06818-f013:**
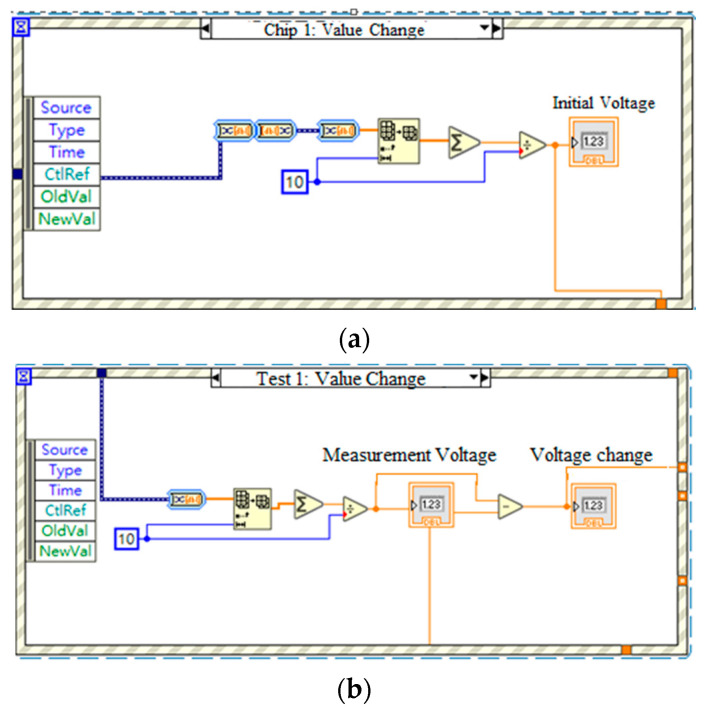
(**a**) Initial voltage measurement code, and (**b**) voltage change calculation code.

**Figure 14 sensors-23-06818-f014:**
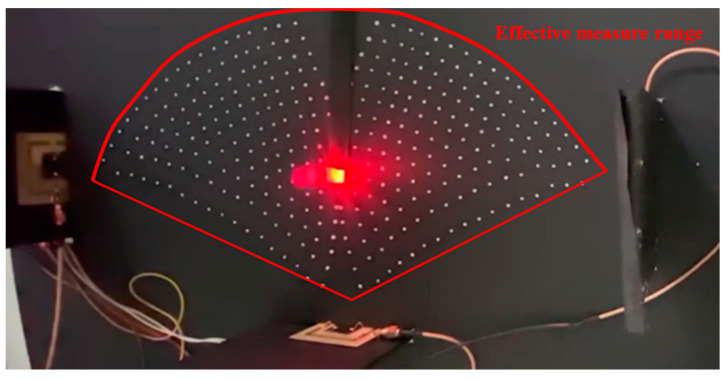
Experimental position measurement system.

**Figure 15 sensors-23-06818-f015:**
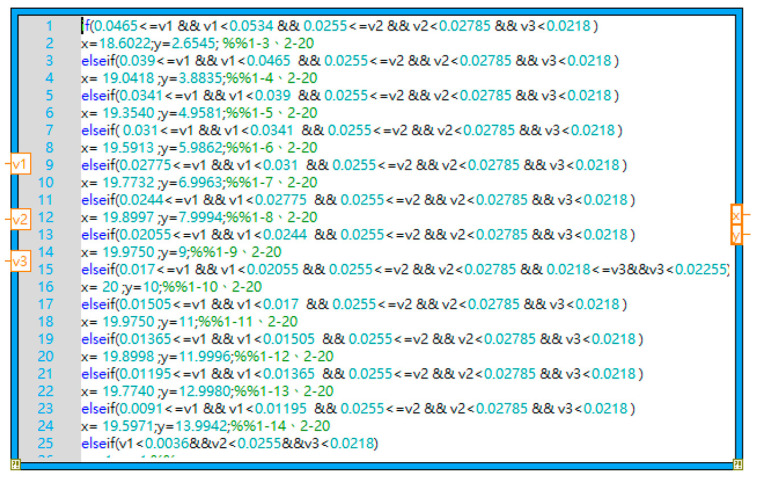
LabVIEW Mathscript position determination code.

**Figure 16 sensors-23-06818-f016:**
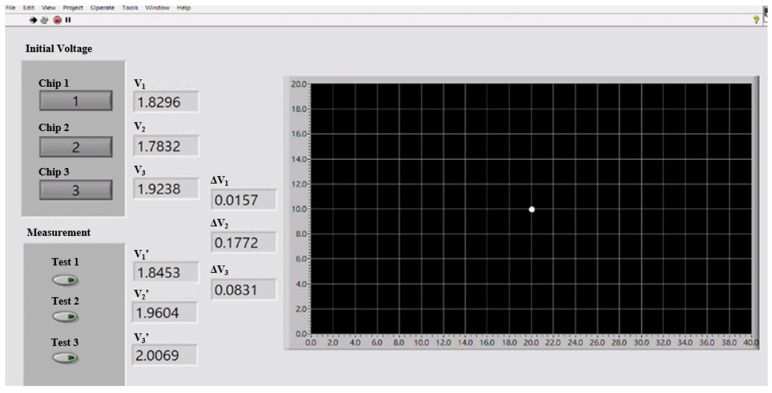
Human–machine interface showing coordinates of sensed target.

**Table 1 sensors-23-06818-t001:** Comparison of measured coordinates and read coordinates of target object.

Point	Measured Coordinate	Read Coordinate	Error
1	X = 19.8, Y = 9.8	X = 20.0, Y = 10.0	X: ±0.2, Y = ±0.2
2	X = 24.1, Y = 9.3	X = 24.0, Y = 9.2	X: ±0.1, Y = ±0.1
3	X = 7.9, Y = 11.7	X = 7.8, Y = 11.8	X: ±0.1, Y = ±0.1
Unit: cm Average Error: ±0.13 (X, Y)			

## Data Availability

The data presented in this study are available on request from the corresponding author.

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
