# Peer review of "Positioning System of Infrared Sensors Based on ZnO Thin Film"

_sensors, 2023, doi:10.3390/s23156818_

Round 1
Reviewer 1 Report
The manuscript sounds good in view of its application and designing and fabrication of the sensing device. However, since it is based on ZnO thin film, I would recommend to add some more characterizations of developed ZnO films specially related to the sensing properties. Thickness various, temperature variation...conductivity, thermal conductivity....That will definitely improve the manuscript and interesting for the readers...
Ok
Reviewer 2 Report
The core of the manuscript is an interesting topic for a broad audience, as it would trigger research from different areas. mY Comments for the paper are:
• Can you provide more details about the fabrication process of the infrared sensors incorporating ZnO pyroelectric films? What were the key parameters and techniques used in the thin-film deposition, photolithography, and etching processes?
• How did the annealing process at 500℃ for 4 hours affect the responsivity of the pyroelectric film? What improvements were observed in the sensor's performance after the annealing step?
• In the experimental evaluation, you mentioned a sensing range of 30 cm. Can you elaborate on how this range was determined, and did you observe any limitations or trade-offs in the sensor's performance at different distances?
• What specific characteristics of the inverse exponential function were observed in the voltage signal variation with distance? How does this function relate to the physical properties of the ZnO pyroelectric film and the thermally-insulated silicon substrate?
• In the positioning system based on three infrared sensors, how were the sensor signals combined to estimate the position accurately? Can you explain the working principle of the LabVIEW implementation and how it accounts for potential errors or uncertainties in the sensor measurements?
• Were there any challenges or limitations encountered during the implementation of the positioning system using infrared sensors? How does the system perform in real-world scenarios with varying environmental conditions or interferences?
• What are some potential applications or use cases for these MEMS-based pyroelectric infrared sensors? Can you discuss how the research findings could be applied in practical settings and industries?
• How do the results and findings presented in this study compare with other existing methods or approaches for positioning systems based on infrared sensors? Can you discuss the advantages and limitations of your proposed system in comparison to alternative techniques?
• In the future development of MEMS-based pyroelectric infrared sensors, what are some areas that could be further improved or optimized to enhance their performance or versatility?
• Considering the growing interest in sensor technology for various applications, what are some potential research directions or areas of exploration that you believe could significantly advance the field of infrared sensors incorporating ZnO pyroelectric films?
Moderate editing of English language required
Reviewer 3 Report
Thank you for a very interesting article. The authors effectively described and presented the topic under discussion. However, I noticed that there is no discussion section included. I believe it would be beneficial to include a section dedicated to discussion. In this section, it would be helpful if the authors could elaborate on how the new knowledge presented in the article can be practically applied. I recommend that the authors clearly articulate the contributions of their paper to both theory, which pertains to the body of conceptual knowledge, and practice, with regards to managers, employees, and policy makers.
Round 2
Reviewer 1 Report
Sorry to say that authors could not understand what were the comments ?
It was mentioned to perform some experiments and study some of the properties related to the sensing ability of the ZnO films which is lacking in the manuscript.
NA
